# microRNAs (miRNAs) in Glioblastoma Multiforme (GBM)—Recent Literature Review

**DOI:** 10.3390/ijms24043521

**Published:** 2023-02-09

**Authors:** Marianna Makowska, Beata Smolarz, Hanna Romanowicz

**Affiliations:** 1Department of Anesthesiology and Operative Intensive Care Medicine, Charité–Universitätsmedizin Berlin, Corporate Member of Freie Universität Berlin, Humboldt-Universität zu Berlin, Augustenburger Platz 1, 13353 Berlin, Germany; 2Laboratory of Cancer Genetics, Department of Pathology, Polish Mother’s Memorial Hospital Research Institute, Rzgowska 281/289, 93-338 Lodz, Poland

**Keywords:** glioblastoma multiforme, miRNA

## Abstract

Glioblastoma multiforme (GBM) is the most common, malignant, poorly promising primary brain tumor. GBM is characterized by an infiltrating growth nature, abundant vascularization, and a rapid and aggressive clinical course. For many years, the standard treatment of gliomas has invariably been surgical treatment supported by radio- and chemotherapy. Due to the location and significant resistance of gliomas to conventional therapies, the prognosis of glioblastoma patients is very poor and the cure rate is low. The search for new therapy targets and effective therapeutic tools for cancer treatment is a current challenge for medicine and science. microRNAs (miRNAs) play a key role in many cellular processes, such as growth, differentiation, cell division, apoptosis, and cell signaling. Their discovery was a breakthrough in the diagnosis and prognosis of many diseases. Understanding the structure of miRNAs may contribute to the understanding of the mechanisms of cellular regulation dependent on miRNA and the pathogenesis of diseases underlying these short non-coding RNAs, including glial brain tumors. This paper provides a detailed review of the latest reports on the relationship between changes in the expression of individual microRNAs and the formation and development of gliomas. The use of miRNAs in the treatment of this cancer is also discussed.

## 1. miRNA Characteristics

Non-coding RNAs are a type of molecule formed as a result of DNA transcription, which do not encode proteins, but perform a variety of structural, enzymatic, and regulatory functions in the cell. Among such RNAs, we distinguish a particularly important group with regulatory functions—microRNAs. So far, it has been confirmed that miRNA are present in all plants and animals, but of the more than the thousands of identified molecules (over 2000 in human cells), only a very small percentage of them have recognized and described functions [1]. It is suspected that about 30% of human genes can be regulated by appropriate miRNAs [2]. After post-transcriptional treatment, miRNA is a single-stranded, short, ~22 nucleotide long, regulatory molecule found in the cytosol of the cell. Together with the corresponding proteins, miRNA forms RNA-inducing silencing complexes (miRISC). Such a complex, thanks to the complementary sequence of nucleotides in miRNA, inhibits the translation of the target mRNA [3]. In this way, the labeled mRNA is degraded by the Ago protein, which prevents further expression of the protein. This inconspicuous molecule has a large impact on the level of gene expression in the cell, and thus, on the functioning of not only a specific cell or tissue, but the whole organism.

miRNA biogenesis is a very complex and varied process between different types of miRNAs (Figure 1) [4]. The differences between the formation of individual molecules begin at the transcription level, among other things, because the sequences encoding miRNAs can be located in intergenic regions or within genes encoding proteins. The primary product of transcription is double-stranded pri-miRNAs (primary miRNA). These are then shortened to 60–70 nucleotide pre-mirRNAs by RNAse III–Drosha endonuclease (forming a complex in mammalian cells with the DGCR8 protein or its homologues in other animals, e.g., Pasha in *Drosophila melanogaster*). Pre-miRNAs are exported outside of the cell nucleus through channels formed in the membrane of the cell nucleus by exportin-5. In the cytoplasm, pre-miRNA is bound by the Dicer endoribonuclease and separated into two strands about 22 nucleotides long [4]. Both threads form a duplex, which consists of an antisense strand (guide) and a sense strand (passenger).The antisense strand, responsible for the suppressive activity of miRNA, is built into the miRNA-induced silencing complex (miRISC). On the other hand, the sense strand usually degrades [5]. The suppression mechanism is based on complementary attachment to regulatory mRNA sequences (including the area at the end of 3′-UTR).

It has been shown that genes for microRNAs are often located in or near fragile sites [6]. Genome damage, such as translocations, deletions, amplifications, and integrations of foreign DNA, for example HPV (human papillomavirus), affect not only the expression of genes that are tumor suppressors (TS), but also the expression of microRNAs. A change in the level of expression of the latter is observed in various types of tumors [7,8,9].

The diversity of genes regulated by miRNA means that in the context of cancer development, these molecules can behave like oncogenes and as transformation suppressors [10]. Based on the level of miRNA expression, normal tissues can be differentiated from cancerous tissues [11,12,13,14]. microRNA expression profiling can be a good diagnostic tool for assessing disease staging or survival and may also be helpful in choosing treatment tailored to the individual needs of the patient [15]. One of the first reports on the participation of miRNA in the tumorigenesis process concerned the effect of miR-17-92 overexpression on the initiation of the carcinogenesis process in mouse lymphatic cells [16]. Similarly, in the case of cells lining the bile ducts, overexpression of miR-29 led to a decrease in the amount of the antiapoptotic factor Mcl-1 and consequently to the cancer transformation of these cells [17]. Suppressor activity was observed in the case of let-7, which is a negative regulator of the expression of oncogenic proteins Ras and c-Myc in human cancer cells of the large intestine [18]. It is now known that deregulation at the level of miRNA expression affects tumors of various origins, including breast [19], colon [20], lung [21], liver [22], and pancreas [23] cancers, as well brain gliomas [24].

In cancer, miRNAs have been shown to modulate cell proliferation and affect invasiveness, angiogenesis, and recurrence [25,26]. miRNA expression disorders are a direct cause of tumor development and are the result of neoplastic processes involving changes in the activity of transcription factors controlling their expression. One of the arguments supporting the influence of changes in miRNA expression on the initiation of the tumorigenesis process indicates the location of their genes on chromosomes near fragile sites, susceptible to deletions, amplifications, point mutations and DNA methylation disorders [27,28]. An example is miR-15a-16, whose gene (located in the 13q14 region) is subject to frequent deletions in chronic lymphocytic leukemia [28,29]. The result of these changes is inhibition of miR-15a and miR-16-1 expression (acting as negative regulators of the anti-apoptotic factor Bcl-2) and, consequently, impaired activation of apoptosis [28].

The main goal of profiling miRNA expression in glioblastoma cells is to identify specific miRNAs whose changes in the level of expression are correlated with the process of tumorigenesis [30]. Currently, in addition to standard techniques (RT-PCR and Q-PCR), high-resolution techniques such as deep sequencing and microarrays are used for these tests. On the basis of microarrays and deep sequencing, a group of microRNAs were selected that undergo significant overexpression in glioma cancer cells compared to control trials, and miRNAs whose levels in glioma cells are reduced [31,32,33]. Table 1 shows typical miRNAs found in glioblastoma multiforme (GBM).

## 2. Glioblastoma Multiforme (GBM)

Gliomas are the most common primary cancers of the central nervous system. They originate from glial cells and account for 40–60% of intracranial tumors. The most common malignant tumor derived from glial tissue is glioblastoma multiforme, accounting for up to 90% of gliomas in adults [46]. Depending on the type of glial cell from which they originate, gliomas are classified as astrocytes (developing from astrocytes), oligodendrocytes (from oligodendrocytes), or linings (from epenymocytes). Due to the degree of biological malignancy, gliomas are divided into low-malignant gliomas with low proliferative potential, slow-growing, minimally invasive gliomas with relatively good prognosis (I and II degree of biological malignancy according to WHO), and high-malignant gliomas characterized by abundant vascularization, intensive proliferation, infiltration of neighboring tissues, and a very poor prognosis (III and IV degree according to WHO) [47]. Glioblastoma multiforme is the most common primary brain tumor, accounts for more than 50% of all gliomas, and has the highest degree of biological malignancy (WHO Grade IV) [47,48,49]. GBM has an infiltrating growth nature, abundant vascularization, and a rapid and aggressive clinical course. GBM is characterized by the occurrence of poorly differentiated neoplastic astrocytic cells, cellular and nuclear atypia, intense mitotic activity, neoangiogenesis, vascular thrombosis, limited apoptosis, and foci of necrosis. Vascular hyperproliferation and necrosis are the basic diagnostic criteria for distinguishing GBM from lower-grade gliomas [50,51]. GBM most often occurs in the supraential part of the cranial cavity—in the frontal, parietal, temporal, and occipital regions—but rarely in the cerebellum [52,53]. Due to the location and significant resistance of gliomas to conventional therapies, the prognosis of patients with glioblastoma is very poor and the cure rate is low [54]. GBM is divided into primary and secondary. The first of these mainly affects elderly patients, while much rarer secondary ones usually affect patients before the age of 45. Primary GBM develops de novo from glial cells, characterized by an aggressive course with a short clinical history of usually less than six months. Secondary multiforme gliomas originate from tumors of the astrocytic series (astrocytics) of the lower degree of WHO malignancy as a result of their transformation and malignancy. Despite differences in their pathogenesis, primary and secondary GBMs are similar in their morphological and clinical terms [55,56]. Neurological symptoms in the course of brain tumors are the consequence of increased intracranial pressure, which is the result of limited possibilities of increasing abnormal mass within the skull, forming a set of symptoms resulting directly from the damage to specific structures in the brain. These symptoms may include headaches, vomiting, disturbances of consciousness, orientation, paresis, changes in personality, mood, and psycho-motor slowdown. 

Often, symptoms depend on the location of the tumor rather than on its histological peculiarities. The absence of specific symptoms makes the cancer difficult to diagnose, and therefore it is usually detected only at an advanced stage of the disease, in most cases using magnetic resonance imaging (MRI). New tools and methods of diagnosis, including magnetic resonance spectroscopy, magnetic resonance imaging and cerebral diffusion imaging, and the use of F18-fluorodeoxyglucose in positron emission tomography, allow for the diagnosis of cancer at an early stage of its development [57,58,59]. For many years, the standard in the treatment of gliomas has been surgical treatment supported by radio- and chemotherapy. Maximum cytoreduction in the tumor (>98% of the tumor) prolongs life up to 9–12 months and also improves the patient’s response to radio- and chemotherapy. However, due to the location of the tumor and its infiltrative nature, surgical intervention is often not possible. For high-beech gliomas (WHO III and IV), radiation therapy (RT) is used as the first adjuvant treatment after surgery. In turn, the standard in chemotherapy is temozolomide (TMZ) and carmustine (gliadel) [60,61]. In the case of relapses, a faster and more aggressive tumor growth is observed in the group of patients treated with TMZ and RT, which additionally shows significant resistance to treatment [62,63].

In recent years, many new promising therapeutic targets and potential therapeutics have been identified. The new approaches are based on tools such as small-molecule inhibitors, monoclonal antibodies, and peptide vaccines used to regulate cell pathways crucial for cancer development, angiogenesis, and to abolish the drug resistance of cancer cells. However, clinical trials did not report the expected results in the form of the effective inhibition of glioma cell proliferation [64,65]. The reasons for these failures are seen in the activation of alternative signaling pathways which bypass the factor turned off by the inhibitor [66]. Although the new approaches initially performed very well, most of them were rejected at the clinical trial stage. Therefore, gliomas invariably remain among the most difficult to treat and are the least promising cancers, with an average survival time of less than a year [67]. In the absence of effective treatments for gliomas and their resistance to treatment, the challenge is to research new therapeutic goals and approaches in the treatment of GBM. The first stage of therapy design is the identification of therapeutic targets, the “shutdown” of which provides a chance to stop pathological processes, including a reduction in cell proliferation and the launching of apoptotic processes. The use of high-throughput DNA sequencing techniques, cDNA microarrays, and proteomic methods provided new knowledge on the pathogenesis of gliomas and allowed the identification of potential therapeutic targets [68,69]. Currently, much attention is focused on transcription factors, extracellular matrix proteins, chaperone proteins, and miRNAs as new promising targets for GBM therapy. The advantage of the latter over the others is their ability to regulate expression at almost every stage. miRNAs can regulate the expression of up to 90% of genes, and consequently affect a number of cellular processes, such as growth, cell differentiation, apoptosis, or cell signaling [70,71]. 

A disturbance in miRNA levels alters the expression of target mRNAs. It is estimated that this may be the cause of over 390 diseases (http://cmbi.bjmu.edu.cn/hmdd, accessed on 1 January 2014), the largest group of these being cancers, including brain tumors.

## 3. miRNAs Expression in GBM

It has been shown that, just as in a healthy brain during its development, the level of individual miRNAs undergoes dynamic changes in a tumor at various stages of its advancement. The miRNA profile in GBM indicates the stage of the disease and can also facilitate the prognosis and selection of appropriate therapy. Based on the level of individual miRNAs, the miRNAs with the highest prognostic value for GBM were selected [72], indicating that that the diagnosis of GBM is also possible on the basis of the analysis of miRNA from the blood and cerebrospinal fluid of patients [73,74,75]. Functional analysis of individual GBM-specific miRNAs indicates that they can act as both oncogenes and tumor suppressors, are responsible for developing resistance to chemotherapy and radiotherapy, stimulate neo-angiogenesis and cell proliferation, and regulate the cell cycle and apoptosis [76,77,78,79,80] (Table 2).

Understanding the GBM-specific miRNA expression profile provides evidence of the involvement of individual miRNAs in the pathogenesis of GBM. This increases the possibility of diagnosing and predicting these cancers. However, we are still far from understanding the mechanisms of cellular regulation involving miRNAs and therefore, also from using miRNAs as potential therapeutic targets. Numerous reports indicate that abnormalities in the expression of selected miRNAs may contribute to the transformation of glial cells and thus to the development of cancer. These include both miRNAs with suppressor and oncogenic functions (Figure 2) [97,98,99,100,101,102,103,104,105,106,107,108,109,110].

## 4. Tumor Suppressor miRNAs

miR-181a and miR-181b are the suppressor molecules whose expression is significantly reduced in glioma cells compared to normal glial cells [111,112]. A decrease in the level of expression of miR-181 genes (miR-181a, 181b, and 181c) was observed in 20–30% of the GBM samples studied [113,114]. It has been shown that the reduction in the pool of miR-181 molecules is proportional to the degree of tumor malignancy (the greatest inhibition of miR-181 expression occurs in stage III and IV tumors). These regulatory factors have been shown to inhibit proliferation, induce apoptosis, and limit cancer cell invasion. Additionally, the induced overexpression of miR-181a and miR-181b in glioma cells results in the loss of the ability of cells to grow independent of contact with the substrate, which is one of the determinants of cell malignancy. Thus, miRNAs from the miR-181 family function as suppressors in glioma cells [115,116].

Reduced expression of miR-34a may also be important in the development of glioma. In glioblastoma cells, the mechanisms that reduce the level of miR-34a probably include epigenetic factors (CpG island methylation disorder) and mutations within the 1p36 locus, which in the case of primary brain tumors, are subject to frequent deletions (70–85%) [117]. This molecule, by regulating the expression of many oncogenes, e.g., *C*-*MET* and *NOTCH,* inhibits the development of cancer [100,118]. It has been proven that transfection of miR-34a cancer cell lines leads to the blockage of cell cycle proliferation and progression and reduces cell survival and invasiveness [119]. One of the direct processes regulated by miR-34a in glioblastoma cells is considered to inhibit the expression of genes of the proteins Notch homolog 1 (Notch1) and Notch homolog 2 (Notch2), which act as transmembrane receptors [120]. With the correct level of miR-34a expression, this mechanism, carried out with the participation of the mesenchymal epithelial transition factor (c-Met), leads to a decrease in signal transduction and subsequently inhibits the process of angiogenesis and proliferation [118]. In GBM cells, elevated levels of Notch1 and 2 receptors stimulate the proliferation and migration of glioma cells by activating the kinases of the AKT-mTOR pathway [121] and also affects the regulation of EGFR receptor expression by the p53 protein, which reduces signal transmission and consequently intensifies the metastasis of cancer cells [30]. Another way to initiate tumor growth with reduced miR-34a expression is to limit the process of inhibition of the transcription of cyclins (E2, D1), kinases (CDK6, CDK4), and Bcl-1, MYC, and E2F3, whose elevated levels in the cell leads to uncontrolled cell cycle progression and the inhibition of apoptosis [122,123]. miR-34a regulates the expression of silent mating type information regulation 2 homolog 1 (SIRT1), the excess of which blocks the process of programmed cell death by binding to the p53 protein [124].

As already stated for cancer stem cells, miR34 plays a visible bimodal role by regulating the Notch and Numb proteins [125]. Numb has been identified as a docking protein involved in the development of Drosophila as an equivalent to Notch, while in various models it works by inducing Notch degradation. In addition, Numb is involved in EGF signaling and internalization of its receptor [126]. Numb also plays a role in stabilizing p53 with a clear implication not only in cancers, but also in stem cells, where p53 has been shown to play a role in stem cell division [127].

miR-34 acts on the regulation of various genes, such as Bcl2, involved in inhibiting the apoptosis pathway, and genes such as NOTCH and NUMB, involved in the development of the nervous system. The effect of miR-34c on NUMB expression can be explained by the interaction on the untranslated 3′ region of NUMB mRNA. This region is conserved among miR-34a and miR-34c. miR34c expression reduces NSC and GSC cell growth and regulates both Bcl2 and NUMB expression. miR-34 clearly inhibits Bcl2, which is involved in resistance to apoptosis, increased cell survival, and response to radiation. Apoptotic resistance can also be affected by NUMB-inducing AKT phosphorylation. miR-34c reduces NUMB expression in both NSC and GSC. miR-34c can inhibit GSC by reducing Bcl2, which could potentially increase the effect of chemotherapy/radiotherapy [128].

miR-34c can be used to treat GBM or other types of cancer. This particular miRNA can be successfully transmitted by viral vectors or extracellular vesicles [129].

The suppressor effect in the development of glioma as well as the migration and invasion of its cells is shown by miR-146b-5p [130,131]. It has been proven that the expression of this miRNA is lower in all types of gliomas than in control astrocytes. miR-146b-5p binds to the 3′UTR region of the *EGFR* gene transcript, inhibiting its translation. The incorporation of this miRNA into cells results in a reduction in the level of protein kinase phosphorylation (AKT) and inhibition of the Pi3K/AKT pathway. This indicates that restoring the normal expression of this miRNA may be helpful in the treatment of invasive forms of cancer [132,133]. Among the genes regulated by miR-146b in GBM, MMP16 has been identified [134]. The MMP16 protein is responsible for the degradation of extracellular matrix components (ECMs), including collagen III, and is specifically expressed in the central nervous system [135]. In GBM cells with a reduced level of miR-146b, increased expression of MMP affects the expansion of the tumor process by supporting the invasion and migration of cancer cells, as well as by the formation of new blood vessels in the tumor environment [136]. 

*EGFR* is also a target gene for miR-7, which is lowered in the expression in glioma cells. It was found that the function of miR-7 associated with the inhibition of the AKT pathway is responsible for limiting the viability and invasiveness of the tumor, which also testifies to the therapeutic potential of this molecule [137,138].

Reduced expression in glioma cancer cells also applies to miR-124, miR-137, and miR-101. miR-124 and miR-137 molecules regulate the expression of the *CDK6* gene and contribute to lowering the level of the CDK6 protein, which is involved in the development of a number of malignant tumors [139,140,141]. This results in the blockade of the cell cycle in the G1 phase and the limitation in the proliferation of glioblastoma multiforme cells, which can be extremely valuable in the treatment of this condition [142,143]. 

The expression of miR-101 varies significantly in glioblastoma cells compared to unchanged cells. Lower levels of this miRNA cause insufficient repression of the enhancer of zeste homolog 2 (EZH2) mRNA translation [73]. This leads to overexpression of methyltransferase EZH2, which induces the proliferation and migration of tumor cells and contributes to the development of tumor vascularization. The level of *EZH2* expression correlates with the survival time of patients [142].

In GBM, reduced expression of miR-128 is observed [61]. High levels of miR-128 have been shown to inhibit glioma cell proliferation in vitro and tumor heterograft growth in vivo by directly regulating the Bmi-1 gene [144]. Mechanically, this effect of miR-128 in GBM was associated with self-renewal inhibition of glioma stem cells (GSC) through the Bmi-1 pathway. miR-128 reduces the proliferation of glioma cells by targeting E2F3a [145,146]. miR-128 inhibits the proliferation, invasion, and self-renewal of GBM and glioma stem cells through the BMI1 and E2F3 pathways [147]. miR-128 has been shown to reduce gliogenesis by down-regulating the EGFR and platelet-derived growth factor receptor alpha (PDGFRA). The targets for miR-128 (except EGFR and PDGFRA) for inhibiting GBM cell proliferation are WEE1, MSI1, and E2F3A [148]. In addition, miR-128 regulates angiogenesis by inhibiting P70S6K1 kinase [149]. The upregulation of miR-128 attenuates the effects of cell proliferation, tumor growth, and angiogenesis [149].

## 5. Onco-miRNA

miR-21 is a miRNA of an oncogenic nature, the overexpression of which is found in many types of cancer, including glioma cells [150,151]. Binding sites for this molecule were found in the 3′UTR regions of transcripts of genes such as programmed cell death 4 (PDCD4), methylthioadenosine phosphorylase (MTAP), and sex-determining region Y box 5 (SOX5). In the development of cancer, PDCD4 is very important, acting as a suppressor gene involved in apoptosis [152]. In the T98G glioma cell line, the level of expression of the *PDCD4* gene shows an inverse relationship with the expression of miR-21, and its reduction leads to the inhibition of the process of apoptosis, which is dependent on this gene [153,154].

miR-10b is highly oncogenic in GBM, suggesting that it may regulate oncogenesis and serve as a useful target in GBM therapy. The overexpression of miR-10b has been found in higher-grade gliomas [155,156]. miR-10b has multiple targets such as RhoC, uPAR, and HOXD10 [157]. By influencing these targets, miR-10b is inhibited, resulting in reduced cell growth, invasion, and angiogenesis, as well as increased apoptosis in GBM [126]. In addition, the direct targets of miR-10b associated with cell growth are BCL2L11, TFAP2C, CDKN1A, and CDKN2A [157]. Inhibition of miR-10b can restore target gene expression and reduce glioblastoma cell growth through apoptosis and/or cell cycle arrest.

Many studies have shown that miR-93 is elevated in GBM [30,158,159,160,161]. miR-93 regulates various glioma cell functions such as proliferation, migration, invasion, cell cycle arrest, and chemoresistance by targeting P21 [162]. miR-93 was shown to control autophagic activity in GSC glioblastoma stem cells by inhibiting BECN1/Beclin 1, ATG5, ATG4B, and SQSTM1/p62 [163]. miR-93 regulates GBM cell viability, tumor growth, and vasculogenesis. In particular, miR-93 enhances the formation of blood vessels by targeting integrin-β 8 [163]. These aspects of miR-93 make this it particularly interesting in the treatment of neo-angiogenesis in GBM.

The increased level of expression in glioblastoma with respect to normal glial cells concerns miR-196 [164,165]. It has been proven that significant overexpression of miR-196 in cancer cells is likely associated with a shorter overall survival of glioblastoma patients [166,167].

Other examples of regulatory molecules whose increased expression is observed in glioma cells and in a number of other cancers include miR-221 and miR-222 [168,169]. The key role of these miRNAs is to control the cell cycle and proliferation by regulating the expression of the P27 and P57 proteins [170,171]. miR-221/222 are also involved in the regulation of apoptosis by directly binding to the 3′UTR mRNA region of the PUMA gene (BCL2 binding component 3), which has recently been recognized as the main mediator of apoptosis, in which the transcription factor TP53 is involved [172,173]. Thus, increased expression of these miRNAs may contribute to inhibiting programmed cell death in glioblastoma [174,175]. Increased activity of miR-221/222 in glioma cells may be due to the improper expression of the transcription factors nuclear factor kappa-light-chain-enhancer of activated B cells (NF-kB) and C-JUN. These factors, binding together to one of the regulatory regions of the miR-221/222 genes, induce their transcription [176,177].

Overexpression of miR-182 increases with the degree of tumor malignancy (a 32-fold increase in miR-182 levels in GBM was observed compared to normal brain tissues) [178,179].

The miR-182 coding sequence was identified in the region of chromosome 7q32.1 within the FRA7H brittle site and the MET gene. MET and FRA7H products have been shown to be frequently amplified in GBM cells [180].

## 6. The Use of miRNAs in GBM Therapy

There are high hopes concerning the use of miRNAs in GBM therapy (Figure 3). miRNAs may become a promising therapeutic target due to their ability to target multiple genes.

The target of therapy may be microRNAs that undergo both elevated and decreased expression in tumors. According to the latest research, two basic strategies for miRNA-based therapy have been proposed, the first of which is to restore the downward-regulated miRNAs of tumor suppressors using microRNA imitators. The second way is to inhibit oncomiR overexpression with microRNA inhibitors.

Tumor suppressor miRNAs have reduced expression in gliomas. In order to normalize their expression profiles, miRNA-based replacement therapies may be used to increase the expression of a given tumor suppressor molecule.

The inhibition of cancer progression is made possible by exogenous oligonucleotides (also known as miRNA-mimicking), which have the same sequence as the corresponding endogenous miRNAs. These oligonucleotides are synthesized and delivered to GBM cells and strongly inhibit tumor growth. A well-known miRNA with reduced expression suppressor function in GBM is miR-34a [150]. In studies in which cell death was induced using miR-34a mimetics in mutant p53, chemically resistant GBM cells, it was shown that miR-34a mimetics can be used as a novel therapeutic agent [181]. In addition, after transfection into U251 glioma cells, miR-203 mimetics significantly reduce the level of phospholipase D2, which is the target of miR-203. This leads to inhibition of proliferation and invasion of U251 cells [182]. Studies of miR-145 and miR-33a in mouse tumors demonstrated their antitumor activity [183].

miRNA inhibitory therapy is used in GBM to inhibit tumor promoting oncomiR. Recently, many mechanisms have been studied, including the use of antisense oligonucleotides. Antisense oligonucleotides (called antagomiR or antimiR) are synthetically produced oligonucleotides that inhibit levels of upward-regulated miRNAs by blocking the interaction between miRNA and its target mRNAs. In the study, antagomirs coupled to the peptide R3V6 were used to inhibit miR-21. It has been shown that R3V6 peptide can serve as an important tool for the delivery of antisense oligonucleotides [184].

The R3V6 peptide protected oligonucleotides from cleavage by nucleases and also increased their delivery. Conjugate has been found to reduce miR-21 expression and promote apoptosis in GBM cells. Antisense anti-miR-21 oligodeoxynocynucleotides were supplied by R3V6 peptide in vivo. It was noted that apoptosis of cancer cells was strongly promoted, which resulted in effective suppression of tumor growth [185]. In studies using 2′-O-methyl (OMe), the antisense oligonucleotide effectively induced apoptosis in GBM by inhibiting the level of miR-21 expression [186].

miRNA sponges are transcripts containing sites that mimic sequences found in the mRNA complementary to the target miRNA. miRNA sponges are longer nucleic acids such as DNA plasmids or transcribed RNA [187]. They inhibit miRNA function, blocking an entire family of related miRNAs [188]. In GBM, the miR-23b sponge inhibits tumor migration, invasion, and progression in vivo [189]. Natural miRNA sponges include circular RNA (circRNA). ciRS-7 and miR-7 have been found to be overexpressed in the brain [190]. The property of reducing further effects of the target miRNA makes miRNA sponges a tool for studying miRNA function in vitro. However, toxicity and side effects can cause an excess of exogenous nucleic acids, reducing the likelihood that miRNA sponges will be successful as therapeutic agents [191].

Viruses are used to effectively deliver miRNAs to cancer cells [192]. Studies using lentiviral vectors to deliver miR-7-3 to U251 cells showed significant inhibition of proliferation and cell cycle arrest [193]. In vitro and in vivo studies, the crispR/cas9 construct was provided to reduce miR-10 expression using a lentiviral vector [194]. Adenovirus-associated viruses (AAVs) are also candidates for delivering miRNAs. For example, AAV-borne miR-26a was systemically delivered to hepatocellular carcinoma (HCC) cells, resulting in cell cycle arrest, increased apoptosis, and reduced tumor growth [195]. These tests may have potential for other cancers, including glioblastoma. However, side effects such as immunotoxicity, inflammatory responses and tissue degeneration induced by immunogenicity, and mutations caused by the inserted sequence are drawbacks which limit the clinical use of viral miRNA [196]. Therefore, non-viral systems may be more suitable for clinical use.

The most successful delivery systems are polymer and lipid nanoparticles, though magnetic nanoparticles have also been used. In GBM, the widely used miRNA carriers are polymer nanoparticles such as poly (lactic-co-glycolic acid) or PLGA and polyethyleneimine (PEI). In order to deliver antimiR-21 and antimiR-10b to GBM cells, PLGA nanoparticles were used. The result was an increase in the sensitivity to TMZ chemotherapy both in vitro and in vivo [197,198,199]. miRNAs can be successfully delivered by PEI nanoparticles [200]. For example, miR-34a encapsulated in PEI nanoparticles has been delivered across the blood-brain barrier as a treatment for GBM [201]. Lipid nanoparticles are a very useful miRNA carrier for clinical applications due to the stability of miRNAs under physiological conditions [202]. As a result of the simultaneous introduction of the antisense oligonucleotide pemetrexed and miR-21 into glioma cells via cationic solid lipid nanoparticles, high cell uptake efficiency with low toxicity has been demonstrated [203].

Studies in mouse models confirm that lipid-based nanoparticle carriers could become a powerful tool for delivering miRNAs and are likely to find a wide clinical application. In studies on mouse models, stable nucleic acid lipid molecules conjugated with chlorotoxin (CTX-conjugated SNALPs) were used. Systemic delivery of anti-miR-21 resulted in reduced proliferation, tumor growth inhibition, and increased apoptosis in a mouse model of GBM [204]. Lipid nanoparticles containing miR-124 have been found to prolong survival, prevent tumor recurrence, and induce immune memory [53,205,206,207,208,209].

miRNAs can be used as new therapeutic approaches in the treatment of glioblastoma multiforme, Alzheimer’s disease (AD), Parkinson’s disease (PD), and other neurodegenerative diseases [210,211]. The improvement of miRNA changes in GBM and neurodegenerative diseases may be helpful in their early detection. Although glioblastoma multiforme and Alzheimer’s disease share the same molecular pathways, there are significant differences in their modulation. Rapid cell proliferation and cell apoptosis arrest are typical features of GBM. In the case of AD, cell damage and subsequent cell death are common consequences. A set of dysregulated 12 miRNAs in both GBM and AD was identified, demonstrating the existence of an inverse relationship between miRNA expression levels in GBM and AD. Three miRNAs were up-regulated in GBM and down-regulated in AD—hsa-miR-106a, hsa-miR-20b and hsa-miR-424)—and 9 were down-regulated in GBM and up-regulated in AD—hsa-miR-1224, hsa-miR-129, hsa-miR-139, hsa-miR-330, hsa-miR-433, hsa-miR-485, hsa-miR-487b, hsa-miR-584, and hsa-miR-885. In addition, hsa-miR-29c was down-regulated in both GBM and AD, suggesting its involvement in both pathologies [210].

Reverse-expressed miRNAs targeting an identical molecule or modulating the same pathway in both GBM and neurodegenerative diseases may provide attractive entry points to a deeper understanding of the underlying molecular physio-pathological mechanism. For example, miRNA-210 targeting brain-derived neurotrophic factor (BDNF), microglia modulating miRNA-21, and miRNA-27a and -132 modulating Tau would help to explain what pathway triggers a neuron to turn into an undifferentiated and immortal cancer cell or broken dying cell. It is also worth noting that miRNA-10b is not expressed in normal brain tissue, so this would provide an attractive diagnostic approach [211].

Extracellular vesicles (EVs) are a heterogeneous population of vesicles released by cells both in vivo and in vitro. They are an extremely important element of information transfer between different cells without requiring their direct contact. EVs are of a high biological importance and are the subject of intensive research. miRNA-transporting exosomes are one of the key elements of intercellular communication in cancer biology. Exosomes play a key role in GBM, Alzheimer’s disease, Parkinson’s disease, epilepsy, and other brain disorders [212]. An interrelation was observed between GBM exosomes and neuronal damage responsible for neuronal disorders. Exosomal miRNAs are present in the body fluids of patients suffering from malignant gliomas [212].

GBM-derived exosomes can increase oxidative stress in cerebellar neurons by reducing cellular antioxidant defenses and increasing oxidative damage [213].

Attention has recently been drawn to the occurrence of subpopulations of stem cells, called “cancer stem cells” (CSCs), in lesions. miRNAs play an important role in CSCs as important regulators of proliferation and differentiation. Several miRNAs are associated with GBM CSC [214]. The most significant are miR-21 and miR-95, which may affect the molecular profiling of GBM and patient survival. Other miRNAs have been identified as potential regulators of CSC immunogenicity. However, further analysis is needed to elucidate the molecular mechanisms behind GBM CSC.

The difficulty in treating glioblastoma multiforme arises from the fact that many microRNAs, including miR-21, miR-34a, miR-135b, and let-7 are associated with chemotherapy and tumor radiation resistance [215]. A significant increase in miR-24-1 and miR-151-5b expression was demonstrated as a result of irradiation of glioblastoma cell lines with doses commonly used in the treatment of brain tumors—2 Gy [215]. Levels of miR-590-3p were elevated in glioblastoma multiforme tissues and radiation-resistant glioblastoma cells. A potential therapeutic target in increasing the radiosensitivity of cancer cells is miR-221/222. The synthesis of CYP3A4, which metabolizes most chemotherapy drugs, including those used to treat gliomas, is increased in brain tumors, and can be inhibited with the participation of miR-148a, -27b and -125b. These miRNAs are designed to reduce glioblastoma chemoresistance. miR-210 is also a promising diagnostic and prognostic biomarker that can be detected in the peripheral blood of glioblastoma patients. Serum miR-210 levels in glioma are significantly elevated. Studying microRNAs circulating in cerebrospinal fluid can help diagnose brain tumors. This is due to the fact that primary brain tumors with a tendency to spread can secrete microRNAs with oncogenic properties that can be detected in the cerebrospinal fluid. Simultaneous testing of the level of expression of miR-15b and miR-21 in the cerebrospinal fluid allows for the differentiation of glioblastoma patients from healthy individuals and CNS lymphoma patients with 90% sensitivity and 100% specificity [31].

## 7. Liquid Biopsy

In the diagnosis of brain tumors, a liquid biopsy may be useful [215]. This is a minimally invasive procedure through which information is obtained from body fluids. This information is similar to what is usually obtained from a tissue biopsy sample. A liquid biopsy can analyze circulating tumor cells (CTCs), circulating DNA-free cells (cfDNA), circulating tumor DNA (ctDNA), circulating cell-free tumor RNA (ctDNA), exosomes, proteins, metabolites, and platelets produced by tumors (TEPs). In the case of glioblastoma multiforme, this material may be derived from tumor tissue and may therefore constitute a genuine and representative sample thereof. The best tested of these include ctDNA and ctRNA.

ctDNA can comprehensively represent the glioblastoma genome image. Depending on its histopathological stage, the rate of detection of ctDNA varies [216,217]. Using the NGS technique, the most common gene mutations (TP53, EGFR, MET, PIK3CA, and NOTCH1, TP53, NF1, EGFR1, MET, APC, and PDGFRA, ERBB2, MET, and EGFR) were selected in the patient’s plasma [216,218]. In patients with glioblastoma, methylation of the MGMT gene promoter was observed in tissues and serum ctDNA [214]. In the case of increased methylation, patients had a better response to treatment with alkylating agents.

A potential diagnostic and prognostic target may be the analysis of miRNA in the serum of patients with glioblastoma. It has been shown that the most significant miRNAs are miR-15b, miR-23a, miR-133a, miR-150, miR-197, miR-497, miR-548b, miR-21, miR-128, miR-342, and miR-205. A change in the expression of these miRNAs has been demonstrated in patients with glioblastoma [219,220,221], and their return to normal levels was observed after surgery and chemotherapy, which indicates their use as biomarkers of response to therapy [220,221].

lncRNAs may be potential diagnostic and prognostic biomarkers in glioma multiforme [220]. Several lncRNAs (HOTAIR, GAS5, H19, and MALAT1) with altered expression levels were detected in blood samples of glioblastoma patients compared with healthy subjects [98,221,222,223]. GAS5 has been shown to be associated with patients’ responses to temozolomide (TMZ) therapy. siRNA, circRNA, snRNA, and snoR-NA may also have potential as biomarkers in the diagnosis and prognosis of glioblastoma [224,225].

In non-invasive diagnostics, exosomal vesicles that are present in body fluids (blood, cerebrospinal fluid, urine) can be used. EVs have a diverse molecular composition (nucleic acids, proteins, lipids, metabolites). These components can be transferred to nearby or distant cells through direct EV contact with the cell membrane, fusion, or internalization [226,227]. Analysis of EVs in the blood, cerebrospinal fluid, or other biological fluid of patients with glioblastoma may have diagnostic and prognostic significance [228,229].

EVs can cross anatomical barriers such as BBBs, which increases their value as a potential biomarker for glioblastoma [230,231,232,233]. In glioblastoma-derived EVs, changes in the EGFRvIII, IDH1, PTEN, and PD-L1 genes were detected [231,233,234]. The release of EVs is affected by the treatment of TMZ. EVs derived from TMZ-resistant patients show increased levels of MGMT expression [235,236] and dysregulated levels of proteins associated with cellular adhesion, such as transglutaminase 2 (TGM2), NESTIN, glycoproteins, CD44, and CD133, which are expressed on the surface of EVs [237]. EVs can therefore serve as tumor biomarkers to monitor TMZ treatment.

TEPs are platelets that have received cancer-related molecules from cancer cells [238]. In glioma, TEPs have been shown to capture tumor-derived EVs with mutant EGFRvIII. The EGFRvIII mutation was detected in 80% of glioblastoma multiforme [239]. RNA derived from TEPs could complement currently used biosources and biomolecules used in the diagnosis of a liquid biopsy. This would improve the early detection of cancer and facilitate non-invasive monitoring of the disease [240]. By analyzing TEPs, it is possible to distinguish cancer patients from healthy ones with an accuracy of 84–96%. In addition, TEP profiling can be used to determine the organ origin of the primary tumor with 71% accuracy. TEP profiles can be differentiated between subtypes of molecular tumors based on EGFR and K-RAS [241].

The presence of CTCs in the patient’s bloodstream, resulting from their separation from the primary tumor, can be used for early diagnosis of the disease, as well as for the selection of a therapy and the monitoring of its effectiveness. The representative presence of CTCs for tumors has been detected in patients with glioblastoma of various stages, including glioblastoma multiforme [242,243]. CTCs derived from glioblastoma exhibited EGFR amplification, which was associated with aggressiveness and with the presence of EGFRvIII [49] and increased expression of SERPINE1, TGFB1, TGFBR2, and VIM genes associated with the mesenchymal subtype [244]. CTCs derived from glioblastoma multiforme have been shown to possess stem cell properties, contributing to the formation of local tumors and relapses [245,246]. Additional elements have been detected in the cerebrospinal fluid of glioblastoma patients that may be components of a liquid biopsy, such as circulating miRNAs and miRNAs derived from EVs [247,248]. These may serve as biomarkers of cerebrospinal fluid for diagnosing and monitoring responses to treatment in patients with glioblastoma [248].

The levels of the nine-miRNA panel (miR-21, miR-218, miR-193b, miR-331, miR374a, miR548c, miR520f, miR27b, and miR-30b) were associated with tumor volume and exhibited a 67% sensitivity and 80% specificity [249]. Elevated levels of miR-21, miR-10b, and miR-15b in cerebrospinal fluid have been shown to be associated with glioma stage, prognosis, and response to treatment [250,251,252]. Cerebrospinal fluid miRNAs have a better diagnostic value, with a higher sensitivity (84%) and specificity (92%) than their serum levels [242].

## 8. Conclusions

Our knowledge of disorders occurring in various types and degrees of glioma malignancy has increased dramatically in recent years [53,207], and many of the changes (confirmed by histopathological analysis) are now complementary to basic diagnostic techniques [207,208].

miRNA studies in cancer processes significantly enrich modern knowledge on the pathogenesis of glioblastoma multiforme cells. miRNAs can be new goals in diagnosis and therapy.

Establishing the miRNA expression profile characteristic of GBM cells is an alternative to obtaining a precise picture of the type and extent of cancerous changes in glioma cells. Experimental verification of high-resolution techniques and in silico analyses provides the chance to obtain a reliable answer to the question of the causes and mechanisms of miRNA disorders in cancer cells. It also creates the possibility of using them as prognostic elements or potential targets in the therapy of brain tumors.

## Figures and Tables

**Figure 1 ijms-24-03521-f001:**
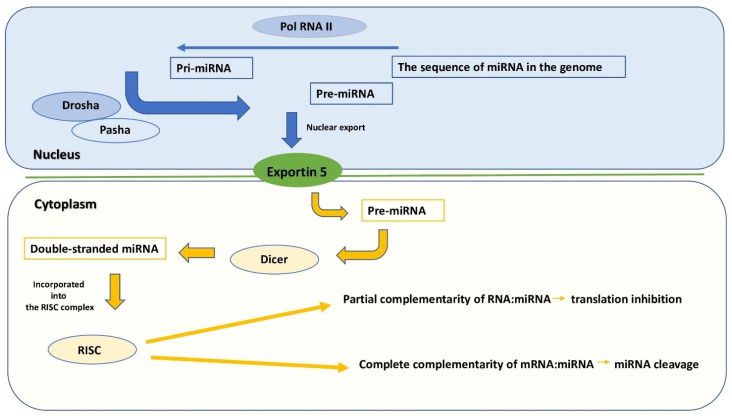
miRNA biogenesis pathway. miRNA genes are transcribed from the sequences encoding them in the genome most often by RNA polymerase II and the resulting pri-miRNA transcripts are subject to modifications of the 5′ (cap) and 3′ (polyadenyl tail) end. They are treated by the Drosha and Pash proteins by cutting out a region with the structure of a hairpin, thanks to which pre-miRNA molecules with a length of about 70 nucleotides are formed. pre-miRNAs are exported from the nucleus by the nuclear transporter exportin 5. In the cytoplasm, pre-miRNA is cut by the Dicer enzyme into double-stranded molecules about 22 nucleotides long. The less thermodynamically stable of the strands is incorporated into the RISC (RNA-induced silencing complex). A high degree of complementarity of the transcript to the RISC-bound miRNA strand results in degradation of the transcript, while partial complementarity inhibits translation.

**Figure 2 ijms-24-03521-f002:**
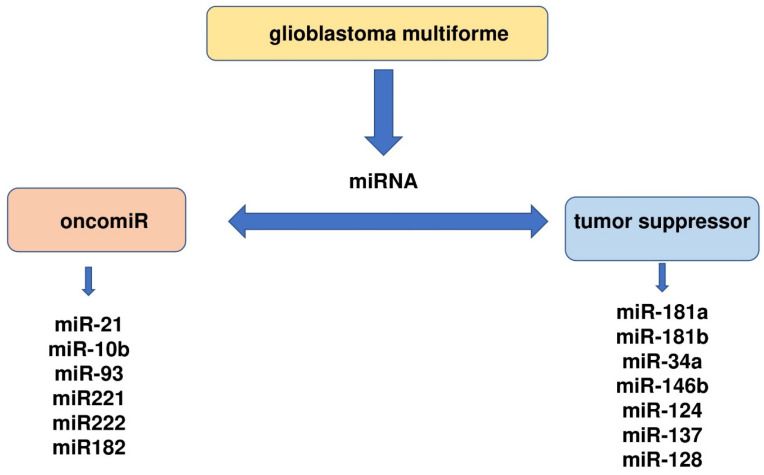
Oncogenic miRNAs and tumor suppressor miRNAs in glioblastoma multiforme.

**Figure 3 ijms-24-03521-f003:**
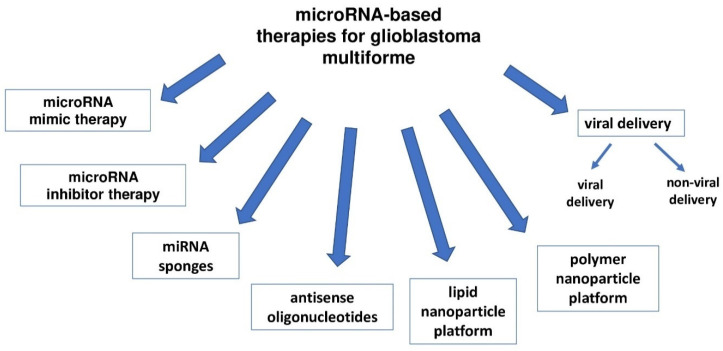
The use of miRNA in therapeutic strategies for glioma.

**Table 1 ijms-24-03521-t001:** miRNAs with both suppressor and oncogenic functions in glioblastoma multiforme.

S. No.	miRNA Name	Expression in Glioblastoma	Role in GBM
1.	miR-21	up-regulated [34]	oncomiR
2.	miR-93	up-regulated [35]	oncomiR
3.	miR-10b	up-regulated [35]	oncomiR
4.	miR-196a	up-regulated [36]	oncomiR
5.	miR-221/222	up-regulated [37]	oncomiR
6.	miR-182	up-regulated [38]	oncomiR
7.	miR-7	down-regulated [39]	tumor suppressor
8.	miR-128	down-regulated [40]	tumor suppressor
9.	miR-124/137	down-regulated [41]	tumor suppressor
10.	miR-101	down-regulated [42]	tumor suppressor
11.	miR-181	down-regulated [40,43]	tumor suppressor
12.	miR-146a	down-regulated [44]	tumor suppressor
13.	miR-137	down-regulated [45]	tumor suppressor
14.	miR-34a	down-regulated [43]	tumor suppressor

**Table 2 ijms-24-03521-t002:** The role of selected miRNAs in GBM.

miRNA-Regulated Process/Pathway	Examples of miRNAs
Growth and differentiation of CSCs (cancer stem cells)	miR-7 [81], miR-9/miR-9 [82], miR10a/miR-10b [83], miR-17-92 [84], miR-124a/miR-137 [85], miR-125a/miR-125b [86], miR-302-367 [86], miR-326 [86]
Cell cycle	miR-21 [87], miR-15b [88], miR-34a [86], miR-221/miR-222 [86]
Proliferation and apoptosis	miR-21, miR-26a [89], miR-101 [90], miR-128 [91], miR-156b-5p, miR-153 [43], miR-181a/miR-181b [92], miR-196a/miR-196b [93], miR-218 [86], miR-381 [86], miR-451 [86], let-7a [86]
Neo-angiogenesis	miR-93, miR-296 [94]
Cell resistance to radio- and chemotherapy	miR-21 [95], miR-125b-2 [86], miR-195 [96], miR-455-3p [96], miR-10a [96]

## Data Availability

Data sharing is not applicable to this article.

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
