# Peer review of "microRNAs (miRNAs) in Glioblastoma Multiforme (GBM)—Recent Literature Review"

_ijms, 2023, doi:10.3390/ijms24043521_

Round 1

Reviewer 1 Report

Makowska and colleagues proposed a well-structured review article on the role of microRNAs in glioblastoma. Overall, almost all the relevant information was provided, however, there are some issues that the authors have to address to further improve their manuscript. Please see the comments below:

1) In Table 2, the authors should provide the references for the miRNAs listed. Same comment for the miRNAs listed in Figure 2;

2) The Chapter entitled “The use of miRNA in GBM therapy” should be revised and improved. In particular, the authors have to add a broad description of the application of miRNAs as diagnostic and prognostic tools in GBM focusing the attention on the possibility of early diagnosis and differential analysis to other neurological diseases with confounding symptoms like Alzheimer’s disease and dementia. In addition, also a description of the detrimental role of GBM towards neurological functions should be added (indicating the mechanisms by which GBM is able to alter brain functions). For this purpose, please see:

- PMID: 31322245

- PMID: 35846075

- PMID: 35883716

- PMID: 26283762

- PMID: 33328891

- PMID: 36203525

- PMID: 33916317

3) The authors should add a new Chapter on the methodologies used for the analysis of miRNAs as well as the sources of samples (liquid biopsy, exosomes, tissue, etc.).

Author Response

Thank you for your review.

I would like to kindly ask you to reconsider the publication of our revised paper:

microRNAs (miRNAs) in Glioblastoma Multiforme (GBM) - Recent Literature Review

I hereby provide responses to the reviewers and list the changes that have been made in the revised version of our paper.

Rev 1

Table 2 has been corrected

Figure 2 has been corrected

Chapter  “The use of miRNA in GBM therapy”

It has been expanded based on additional suggested literature

Added ‘liquid biopsy’ chapter

I hope you find our revised Manuscript satisfying so that it can meet the criteria of publication in your Journal.

Looking forward to hearing from you,

Yours sincerely,

Beata Smolarz

Reviewer 2 Report

Major points

The manuscriptmicroRNAs (miRNAs) in Glioblastoma Multiforme (GBM) - Recent Literature Review”, however from my point of view the manuscript needs professional proofreading editing for the numerous flaws in the text.

Minor Points:

In addition to the flaws, there are some sentences not strictly accurate i.e.: “The use of miRNA in the treatment of this cancer was also discussed” this must be rewritten inThe possible use of miRNA in the treatment of this cancer was also discussed”. No clinical trials have been mentioned or started.

It is mentioned miR34a, however, miR34c effect is not considered in GBM [PMID: 30890698, PMID: 34681654].

Page 7: miR34a affects EGFR pathway and p53  “It also affects the regulation of EGFR receptor expression by the p53 protein, which reduces signal transmission and consequently intensifies the metastasis of cancer cells”. Must be considered that miR34a affects Numb (PMID: 29511160 that is involved in EGFR signaling (PMID: 11121447) and in regulation and expression of p53/p73 (PMID: 18172499, PMID: 30890698).

Check the references (i.e. #46 must be substituted with a more recent issue).

Author Response

Thank you for your review.

I would like to kindly ask you to reconsider the publication of our revised paper:

microRNAs (miRNAs) in Glioblastoma Multiforme (GBM) - Recent Literature Review

I hereby provide responses to the reviewers and list the changes that have been made in the revised version of our paper.

We have followed the valuable comments, for which we thank you. Corrections are placed on page 7 (in red font). The article has been extended with additional literature data in the following chapters ‘The use of miRNA in GBM therapy’ and ‘liquid biopsy’

I hope you find our revised Manuscript satisfying so that it can meet the criteria of publication in your Journal.

Looking forward to hearing from you,

Yours sincerely,

Beata Smolarz

Reviewer 3 Report

Many relevant recent references are missing and no clear and innovative perspective is provided.

Author Response

Thank you for your review.

I would like to kindly ask you to reconsider the publication of our revised paper:

microRNAs (miRNAs) in Glioblastoma Multiforme (GBM) - Recent Literature Review

I hereby provide responses to the reviewers and list the changes that have been made in the revised version of our paper.

The article has been extended with additional literature data in the following chapters ‘The use of miRNA in GBM therapy’ and ‘liquid biopsy’

I hope you find our revised Manuscript satisfying so that it can meet the criteria of publication in your Journal.

Looking forward to hearing from you,

Yours sincerely,

Beata Smolarz

Round 2

Reviewer 2 Report

some flaws are still present in the paper, professional editing from the authors or the journal could be useful

Author Response

Thank you for your review. We checked the article again.

It would be easier for us if we knew what specific flaws need to be corrected. We want to emphasize that we have added the latest information to the article (chapters Liquid biopsy, The use of miRNA in GBM therapy)

In addition, we have expanded the literature.

We reviewed the article, tried to correct all typos, punctuation, etc. Other reviewers had no comments on our work, so please evaluate it positively.

Best Regards

Beata Smolarz